# Weighted Gene Co-Expression Network Analysis (WGCNA) Discovered Novel Long Non-Coding RNAs for Polycystic Ovary Syndrome

**DOI:** 10.3390/biomedicines11020518

**Published:** 2023-02-10

**Authors:** Roozbeh Heidarzadehpilehrood, Maryam Pirhoushiaran, Malina Binti Osman, Habibah Abdul Hamid, King-Hwa Ling

**Affiliations:** 1Department of Obstetrics & Gynaecology, Faculty of Medicine and Health Sciences, Universiti Putra Malaysia, Serdang 43400, Malaysia; 2Department of Medical Genetics, School of Medicine, Tehran University of Medical Sciences, Tehran 1417613151, Iran; 3Department of Medical Microbiology, Faculty of Medicine and Health Sciences, Universiti Putra Malaysia, Serdang 43400, Malaysia; 4Department of Biomedical Science, Faculty of Medicine and Health Sciences, Universiti Putra Malaysia, Serdang 43400, Malaysia

**Keywords:** PCOS, lncRNA, bioinformatics, WGCNA, module, hub gene

## Abstract

Polycystic ovary syndrome (PCOS) affects reproductive-age women. This condition causes infertility, insulin resistance, obesity, and heart difficulties. The molecular basis and mechanism of PCOS might potentially generate effective treatments. Long non-coding RNAs (lncRNAs) show control over multifactorial disorders’ growth and incidence. Numerous studies have emphasized its significance and alterations in PCOS. We used bioinformatic methods to find novel dysregulated lncRNAs in PCOS. To achieve this objective, the gene expression profile of GSE48301, comprising PCOS patients and normal control tissue samples, was evaluated using the R limma package with the following cut-off criterion: *p*-value < 0.05. Firstly, weighted gene co-expression network analysis (WGCNA) was used to determine the co-expression genes of lncRNAs; subsequently, hub gene identification and pathway enrichment analysis were used. With the defined criteria, nine novel dysregulated lncRNAs were identified. In WGCNA, different colors represent different modules. In the current study, WGCNA resulted in turquoise, gray, blue, and black co-expression modules with dysregulated lncRNAs. The pathway enrichment analysis of these co-expressed modules revealed enrichment in PCOS-associated pathways, including gene expression, signal transduction, metabolism, and apoptosis. In addition, *CCT7*, *EFTUD2*, *ESR1*, *JUN*, *NDUFAB1*, *CTTNB1*, *GRB2*, and *CTNNB1* were identified as hub genes, and some of them have been investigated in PCOS. This study uncovered nine novel PCOS-related lncRNAs. To confirm how these lncRNAs control translational modification in PCOS, functional studies are required.

## 1. Introduction

Polycystic ovary syndrome (PCOS) is the most prevalent endocrine/metabolic disease in women of reproductive age, affecting 5–26% of women globally (depending on the diagnostic criteria used) [1,2]. PCOS has been linked to various disorders, including metabolic syndrome, cardiovascular disease, Type 2 diabetes, hypertension, and ovulatory infertility [3,4,5]. Four distinct subtypes of PCOS exist: inflammatory, hidden cause, pill-induced, and insulin-resistant (the most common subtype) [6,7]. Although the precise cause of the condition is unknown, genetic, epigenetic, environmental, and behavioral variables have all been linked to PCOS. In clinical and/or biochemical contexts, its diagnosis is based on the presence of at least two of the three primary symptoms, which include ovarian cysts, elevated testosterone levels, and irregular periods. Despite differing views on the current PCOS diagnostic criteria, a professional opinion based on an ultrasound, a pelvic exam, and blood tests may establish the diagnosis [8]. Although the current blood tests for PCOS diagnosis rely mostly on hormone levels and endocrine function, there is an increasing clinical need to identify sensitive and specific genetic biomarkers in tissue and blood specimens [9]. Biomarker research may also help physicians make early diagnoses, molecularly categorize diseases in the clinic, and provide information on the underlying molecular processes of disorders and their derivatives, most notably cancer and diabetes [10]. Similar to treating cancer and common multifactorial disorders such as diabetes, treating PCOS requires a thorough examination of the underlying molecular mechanisms, the identification of novel and specific genetic biomarkers, and the modification of conventional therapeutic options. In silico and bioinformatic studies now provide a plethora of research tools for achieving these objectives. LncRNAs are a subset of non-coding RNAs (ncRNAs) that range in length from around 200 nucleotides to several kilobases and have significant tissue specialization [11]. The function of a lncRNA is dependent on its distinct subcellular localization [12]. The nucleoplasmic lncRNAs engage in gene regulation at the epigenetic and transcriptional levels, including histone modifications [13], DNA methylation [14], and chromatin remodeling [15]. Nuclear lncRNAs also interact with chromatin modification complexes [16], transcription factors [17], and nucleoplasmic proteins [18]. Meanwhile, lncRNAs in the cytoplasm engage in gene regulation at the post-transcriptional and translational stages, including interactions with proteins in the cytoplasm [19], metabolic regulation of mRNA [20,21], and interactions with microRNAs [22,23].

There is mounting evidence that lncRNA plays a significant role in developing human disorders [24,25]. PCOS is among the conditions in which lncRNA dysfunctions may play a role in the disease’s state and progression. In this respect, Li et al. indicated that lncRNA *GAS5* was increased in PCOS and may contribute to development of the disease by influencing cell death and the production of IL-6 [26]. Qin et al. demonstrated for the first time that lncRNA *H19* was related to PCOS, making it a promising diagnostic factor for early endocrine and metabolic problems in PCOS [27]. Liu et al. discovered a correlation between the expression of lncRNA-*Xist* and the pathophysiology of PCOS. In Liu et al.’s study, the expression of lncRNA-*Xist* was downregulated, resulting in increased cancer cell growth and migration [28]. Given the importance of lncRNAs in PCOS, it is essential to identify new lncRNAs and their involvement in the disease’s pathophysiology. In recent decades, massive efforts have been made, both experimentally and in silico, to better comprehend the function of lncRNAs and the possibility of using them as genetic biomarkers for early diagnosis, treatment options, and prognosis. This bioinformatic study aimed to identify lncRNAs that play a crucial role in PCOS.

Genes that are a component of the same signaling and metabolic pathway or have comparable roles tend to be expressed together under various circumstances [29]. Networks of gene sets (i.e., modules) whose expression is strongly correlated are created by using co-expression gene module analysis. WGCNA is one of the most often used methods for identifying co-expressed gene modules. It is a prevalent systems biology method used not only to identify co-expressed gene modules but also to detect the central players (i.e., hub genes) within the modules. It provides a chance to carry out a higher-resolution analysis that may better identify the most significant functional genes that may give a more robust bio-signature for phenotypic features, resulting in more appropriate biomarker candidates for future investigations. The results of WGCNA are shown in different colors; different colors represent different co-expressed gene modules [30]. In the current study, GSE48301 was analyzed using the WGCNA technique in R software, resulting in the identification of nine novel differentially expressed lncRNAs (DElnc) between PCOS patients and normal controls. The biological pathway analyses and interaction networks are displayed to obtain a deeper understanding of the identification of DElnc co-expression gene modules.

## 2. Materials and Methods

### 2.1. PCOS Data Acquisition

To discover candidate lncRNA, PCOS microarray data with the dataset number GSE48301 were attained from the Gene Expression Omnibus (GEO) database (https://www.ncbi.nlm.nih.gov/geo/query/acc.cgi?acc=gse48301, accessed on 26 July 2018). GSE48301 is provided by the University of California, San Francisco, CA, USA. GSE48301 contains 29 samples (14 PCOS patient samples and 15 normal control samples). These 29 samples were from cells isolated by fluorescence-activated cell sorting (FACS) from four different endometrial cell populations, including epithelial cells (eEP), endothelial cells (eEN), stromal fibroblasts (eSF), and mesenchymal stem cells (eMSC). The platform of the identified dataset was the [HuGene-1_0-st] Affymetrix Human Gene 1.0 ST Array.

### 2.2. Processing of the Dataset and Identification of (Differential Expression Genes) DEGs

The GEOquery package was used to download and examine the expression profile data in the R environment [31]. Through use of the Robust Multi-array Average (RMA) method in the affy package of Bioconductor “http://www.bioconductor.org (version 3.16, released on 2 November 2022)”, background correction, quantile normalization, and pro-summarizing were conducted on the gene expression profile data.

To eliminate false positives and maintain high expression levels of the DEGs for further analysis, we only retained the probes that were “present (P)” in more than 50% of all the samples in the dataset, as determined by the mas5calls function in the affy package [32]. Through use of the Limma (linear models for microarray data) package, the DEGs were analyzed [33]. After the *t*-test, genes with a *p*-value < 0.05 were selected as DEGs. The top DElncs were identified on the basis of the same defined criteria.

### 2.3. WGCNA and Module Identification

Given that the majority of lncRNAs have unknown activities, the identification of their functions is strongly dependent on the assessment of their co-expressed genes. To determine the comparative relevance of lncRNAs and potential module involvement, a network analysis was accomplished using the WGCNA package in R [34]. In brief, WGCNA was conducted on the GSE48301 dataset, which included 14 PCOS patients and 15 healthy controls. Co-expression networks were generated using a soft threshold power to distinguish modules with diverse expression patterns. The Pearson correlation coefficient was then used to evaluate the weighted co-expression connections included within the adjacency matrix. The co-expression relationships between genes were then calculated using a similarity function for topological overlap matrices. The networks were constructed by clustering genes with extremely similar co-expression patterns. Thus, the modules were obtained, including the required important lncRNAs and their co-expressed modular genes.

### 2.4. Enrichment Analysis of the Modules and Identification of the Interaction Networks and Hub Genes

Funrich “http://funrich.org/faq (version 3.1.4, released on 12 February 2020)”, a powerful tool for gene and protein functional analysis, was utilized to discover and enrich the identified co-expressed lncRNA gene modules [35]. The *p*-value in the Funrich enrichment analysis was set at <0.05. The Search Tool for Interacting Genes (STRING) database “https://string-db.org/ (version 11.5, accessed on 6 September 2022)”, is a web-based evaluation method for protein protein interaction (PPI) data [36]. STRING was applied to the relevant modules, and only interactions with a total score higher than 0.90 were deemed to be significant. Through use of Cytoscape software, the PPI network, including the critical gene pairs, was then displayed [37]. In addition, the top 10 hub genes were identified using the cytohubba plug-in in the PPI network [38].

## 3. Results

### 3.1. Identification of Differentially Expressed lncRNAs

The purpose of this DEG analysis was to identify the DElncs that had the greatest changes in expression. Nine DElncs were determined on the basis of the defined criterion (*p*-value < 0.05) and are detailed in Table 1, Figure 1 illustrates the box plot and volcano plot of the DEGs in the PCOS samples compared with the normal samples.

### 3.2. Analysis of Weighted Gene Co-Expression Networks and Classification of the Desired Module

In this research, a network of co-expressed genes was established using GSE48301, and the expression levels of 2808 DEGs were examined using the “WGCNA” package used to build the co-expression network. The “hclust” R function was used to conduct the hierarchical clustering analysis, which mainly included the removal of outliers. Meanwhile, the function of pickSoftThreshold was utilized to establish scale independence and analyze the mean connectivity of the modules with various power levels. Next, to ensure a scale-free network, we used β = 12 as the soft thresholding power (Figure 2a) and used a linear regression plot to confirm the scale-free topology R2 (scale-free R2 = 0.93). (Figure 2a). Therefore, β = 12 was used to generate a hierarchical clustering tree including various colors representing distinct modules. A topological overlap matrix (TOM)was generated using the adjacency and correlation matrices of the gene expression profile. As shown in Figure 2b, four co-expressed gene modules were discovered, each of which was color-coded. In conclusion, we investigated the dynamic connections among the four modules, designed the network’s heatmap, and displayed the relative independence of each module (Figure 3). Among the modules obtained, four modules were identified that were co-expressed with dysregulated lncRNAs, as shown in Table 1.

### 3.3. Pathway Enrichment and Establishment of the PPI Network and Hub Genes

Co-expression modules with the identified DElncs in the biological pathway enrichment analysis were enriched in multiple pathways (Figure 4), as summarized in Table 2. To explore the biological characteristics of the desired modules more deeply, PPI networks were created using the STRING database (Figure 5). In addition, the module’s PPI characteristics, including the number of nodes and edges, and subsequently, utilizing cytohubba, hub genes were screened, as presented in detail in Table 1.

## 4. Discussion

PCOS is a diverse endocrine disorder with a high incidence in and socioeconomic implications for women of reproductive age [39,40]. Extended excess estrogen or a deficiency in progesterone results in atypical endometrial hyperplasia [41]. Women with PCOS with endometrial hyperplasia have a greater chance of developing endometrial cancer than women without PCOS [42]. In addition, women with PCOS have an increased risk of obstetric problems such as preeclampsia, gestational diabetes, and premature delivery [43]. Numerous genetic variables have been previously linked to the development of PCOS [44]. To the best of our knowledge, however, the role of lncRNAs in PCOS has not yet been thoroughly investigated. The exact levels of lncRNA expression are crucial because their expression levels are relatively low compared with genes that produce proteins and are significantly less expressed than proteins. Since lncRNAs are only expressed in specific tissues and have a relatively low abundance overall, their expression must be tightly controlled [45]. LncRNAs are involved in controlling the expression of other genes, and small alterations in their expression may significantly affect the expression of other genes. Therefore, dysregulated lncRNAs disrupt the network, according to the co-expression of the lncRNAs and mRNAs [46]. In this study, we tried to find novel dysregulated lncRNAs with the potential for developing PCOS by using a dataset that included PCOS patients’ samples compared with normal controls.

As a result of the analysis, nine dysregulated lncRNAs were obtained, of which *LOC102725104*, *LINC00328*, *LOC102725070*, *LOC101059935*, and *LOC100127951* showed increased expression levels, whereas *LOC648987*, *LOC93622*, *LOC644936*, and *LINC00998* lncRNAs showed decreased expression levels. The characteristics of these lncRNAs are shown in Table 1. To the best of our knowledge, some of these lncRNAs have been evaluated in other diseases; however, none of these lncRNAs have been studied in PCOS. The majority of these lncRNAs are new and have been recently investigated in other diseases, and a limited number of studies is available on them.

In our study, the upregulation of *LOC102725104* was observed in PCOS patients compared with healthy controls. Xunnan Zhang et al. discovered the downregulation of this molecule in the blood of patients with ST-segment elevation myocardial infarction (STEMI) compared with stable coronary artery disease (CAD) [47], which was in contrast to our research. As in our study, no functional study of *LOC102725104* was performed in their research [47]. Hence, performing functional analyses may provide key evidence to understand the discrepancy between our results and those of Xunnan Zhang et al. [47]. In the current research, upregulation of *LINC00328* was observed in PCOS patients compared with healthy controls. Interestingly, *LINC00328* was significantly increased in the blood of those with Parkinson’s disease [48,49] compared with the control group. It also plays an important role in the competitive endogenous RNA (CeRNA) network in Parkinson’s disease via different pathways, especially those enriched in the gonadotropin-releasing hormone (GnRH), insulin, and MAPK signaling pathways [50]. These pathways (GnRH, insulin, and MAPK signaling) are among the most important pathways involved in the pathophysiology of PCOS. Hence, we hypothesized that *LINC00328* might be an important molecule in the neuronal–reproductive–metabolic axis in the pathophysiology of PCOS. *LOC648987* and *LOC93622* were downregulated in our research and were allocated to the grey co-expression module. Some pathway analyses of the grey module indicated enrichment in the metabolic pathways, including respiratory electron transport, ATP synthesis by chemiosmotic coupling, the citric acid (TCA) cycle, and respiratory electron transport. Obesity is frequent in women with PCOS, and obese people are thought to have lower energy expenditure (EE) rates than non-obese people [51]. Hence, the decreased expression of *LOC648987* and *LOC93622* might be due to reduced energy expenditure in PCOS patients compared with the controls. Downregulation of *LINC00998* was observed in our study, which was in line with the results of Cai et al. [52] and Fang et al. [53]. In Cai et al.’s research, downregulation of *LINC00998* increased malignant phenotypes in the glioma cells and was associated with a dismal prognosis in glioma patients [52]. Fang et al. found that *LINC00998* was considerably reduced in human AML, and was associated with recurrence and poor prognosis [53]. Furthermore, the expression of *LINC00998* was significantly lower in patients with major depressive disorder (MDD) than in controls, which was consistent with our microarray data [54]. Downregulation of *LOC644936* was observed in our study, which was in contrast with the findings of Jurgec et al. [55]. According to their results, *LOC644936* was identified as a novel non-coding RNA for acute myeloid leukemia (AML) and chronic myeloid leukemia (CML), and was upregulated. Hence, a functional analysis of *LOC644936* might be an experimental way to assess the exact role of this molecule.

To better understand these lncRNAs, an analysis of their co-expression module genes’ pathways was conducted, and the modules’ hub genes were identified. The term “gene module” refers to a cluster of densely related genes in terms of their level of co-expression [56]. WGCNA identifies gene modules using hierarchical clustering and color to signify modules; genes not allocated to any modules are placed in a gray module [56]. Four dysregulated lncRNAs, including *LOC102725104*, *LINC00328*, *LOC102725070*, and *LOC101059935*, which had increased expression, belonged to the turquoise module. Pathway analysis of the turquoise module indicated that it was mainly enriched in terms of gene expression, HIV infection, the host interactions of HIV factors, and Class I MHC-mediated antigen processing and presentation. The process of gene expression is one of the critical pathways involved in the most basic activities of the cell. The complicated regulation of gene expression comprises interactions among DNA, RNA, proteins, and the environment [57]. LncRNAs serve as regulators of gene expression, and the co-expression of these dysregulated lncRNAs with PCOS-associated genes demonstrated the significance of these dysregulated lncRNAs. In addition, there is a lack of information on the incidence of PCOS symptoms in HIV-positive women, such as dysfunctional menstruation, a rise in the number of ovarian follicles, and hormonal imbalances [58]. Alternatively, several of the turquoise module’s hub genes have been well documented. In this regard, insulin resistance in the skeletal muscle is a major contributor to the onset of T2D in PCOS-affected women [59]. Nilsson et al. found that *CCT7* is one of the genes with the most prominent differential expression in the skeletal muscle of women with PCOS compared with controls [60]. Similar to our findings, Hou et al. included *EFTUD2* in PCOS as a hub gene in their investigation of protein–protein interactions [61]. EFTUD2 is a spliceosomal GTPase with a wide range of biological roles, including developmental abnormalities [62], spliceosome activation [63], and immunological responses [64,65]. It is important to note that these pathways and hub genes could play a major role in PCOS and should be looked into further.

The gray module included genes co-expressed with the dysregulated lncRNAs *LOC100127951*, LOC648987, and LOC93622, of which the expression levels of *LOC100127951* increased while the expression levels of *LOC648987* and *LOC93622* significantly decreased. Pathway analysis of the gray module revealed pathways for signal transduction, GPCR signaling, GPCR downstream signaling, and olfactory signaling to be significantly enriched. In this regard, critical signaling pathways and transduction can be affected in PCOS, including the TNF signaling [66], androgen receptor (AR) signaling [1], cAMP signal transduction [67], estrogen signal transduction [68], insulin signaling [69], MAPK signaling [70], and WNT/β-catenin signaling pathways [71]. Notably, the signaling pathway of the G-protein-coupled receptor (GPCR) protein serves as one of the key pathways of receptor-linked cell surface signal transduction [72]. This system is directly connected to the insulin receptor signaling pathway and to many additional signaling pathways that affect metabolic functions and inflammation [4]. Currently, researchers have focused mostly on genes connected with the hypothalamic–pituitary–adrenal axis while examining the mechanism of GPCR in PCOS. Kisspeptin, which is encoded by the *Kiss1* gene, controls the reproductive endocrine axis, which includes the hypothalamus, anterior pituitary, and gonads. Kisspeptin’s activity is carried out by GPCR 54, which is located on the nerve cells which produce GnRH. The researchers discovered that non-obese PCOS women regularly have an anomaly in their hypothalamic GnRH pulse rate, which functions on GnRH receptors in the pituitary gland, resulting in excessive LH secretion and inadequate FSH secretion, and a rise in androgen production in ovarian theca cells. Consequently, this will alter the growth of follicles and lessen the effect of progesterone on GnRH secretion by lowering the level of progesterone, which will increase the frequency of GnRH pulses [73]. Remarkably, some gray module hub genes have been studied in PCOS, including *ESR1*, *JUN*, and *TPI1*. Earlier genetically designed experiments on knockout mice have shown a fascinating association between the absence of *ESR1* and the PCOS symptoms of intermittent estrus, infertility, elevated androgen levels, and hemorrhagic follicle formation [74]. In addition, Liu et al. observed that the *JUN*-mediated cell cycle pathway was enhanced three- to fourfold in PCOS [75]. In contrast, Corton et al. discovered the downregulation of *TPI* in PCOS patients’ adipose tissue, which may have led to disruption of the cytoskeleton [76]. TPI1, an enzyme that facilitates the interconversion of two major compounds in the preliminary glycolysis phase, interferes with Rho in the regulation of intracellular sodium, possibly through the activation of Na,K-ATPase by providing the glycolytic ATP that provides energy for membrane activities. This interaction occurs because TPI1 catalyzes the completion of the preparatory phase of glycolysis [77]. In addition, TPI1 communicates indirectly with structural proteins such as actin and microtubules through its association with the plasma membrane [77]. Consequently, the downregulation of *TPI1* could potentially impair several functions of adipose tissue cells. *TPI1*, one of the genes responsible for increased glycolysis, is overexpressed in PCOS patients’ Th cells, according to an earlier study [78].

The blue and black co-expressed gene modules include the downregulated lncRNAs *LINC00998* and *LOC644936*, respectively. Among the enriched pathways of these co-expressed modules, metabolism and the metabolism of RNA were related to the blue module, and apoptosis was related to the black module. These were the enrichments with the most significant number of co-expressed genes. These pathways are among the fundamental pathways that are involved in the pathophysiology of PCOS. Notably, some of the blue and black modules’ hub genes have been studied in PCOS, including *NDUFAB1*, *GRB2*, and *CTNNB1*. *NDUFAB1*, also known as NADH: Ubiquinone Oxidoreductase Subunit AB1, is a mitochondrial acyl carrier protein that functions in lipid metabolism by interacting with other mitochondrial proteins [79]. It is a critical component in the production of fatty acids in the mitochondria [80], and it produces lipoic acid, a cofactor required by many mitochondrial enzymes [81]. The growth factor receptor-bound protein-2 (GRB2) is an adapter protein that is required for cellular functioning. Depending on whether it is activated or inhibited, it may either stimulate or hinder cellular transformation and proliferation. It is essential for connecting the cell surface growth receptors (EGFR) to the Ras signaling pathway [82]. In the research conducted by Corbould et al., the phosphorylation of p38 mitogen-activated protein kinase and the signal from the insulin receptor to GRB2 were shown to be similar in PCOS patients and controls [83]. Furthermore, *CTNNB1* encodes catenin, which acts as a gene expression regulator in the WNT signaling pathway and is required for cell–cell adhesion [84]. Compared with the control group, granulosa cells from PCOS patients demonstrated a substantial drop in the expression of *CTNNB1* [71]. Similarly, Zhang et al. identified *CTNNB1* as a candidate gene in PCOS [85], and its upregulation in women with PCOS relative to the controls was revealed [86].

Our study has the following main limitations: we confirmed our results using data from a publicly accessible database, but we did not assess the expression of the linked genes or the underlying biological processes. Moreover, additional high-quality biological research with greater sample numbers must be conducted to confirm our results.

## 5. Conclusions

Our PCOS research resulted in the discovery of nine new dysregulated lncRNAs. WGCNA analysis discovered the co-expression gene modules of these lncRNAs, and we subjected them to biological pathway enrichment analysis in order to uncover as many of them possible. The hub genes found in the PPI network demonstrated the significance of these genes in developing PCOS. These findings are preliminary, and more in vitro and in vivo research may reinforce them. While the possible significance of these lncRNAs at the transcript level deserves additional exploration, these findings provide the framework for future research into the effects of lncRNAs on PCOS.

## Figures and Tables

**Figure 1 biomedicines-11-00518-f001:**
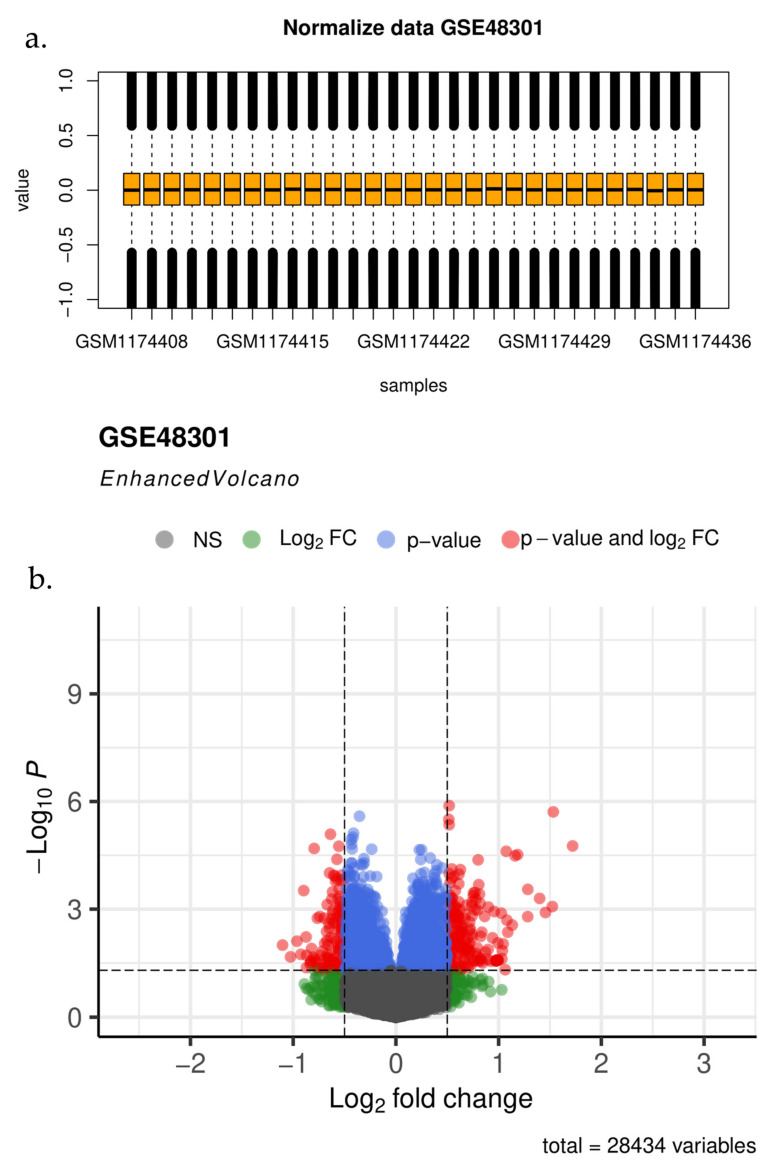
Boxplot of the datasets and volcano plot of the DEGs. (**a**) In the box plot, distinct arrays had identical expression level medians, indicating correct adjustments. (**b**) The x-axis shows the log fold change, and the y-axis shows the −log_10_ (adjusted *p*-value). The gray dots show genes with no significant difference. Screening for DEGs was carried out with a *p*-value < 0.05.

**Figure 2 biomedicines-11-00518-f002:**
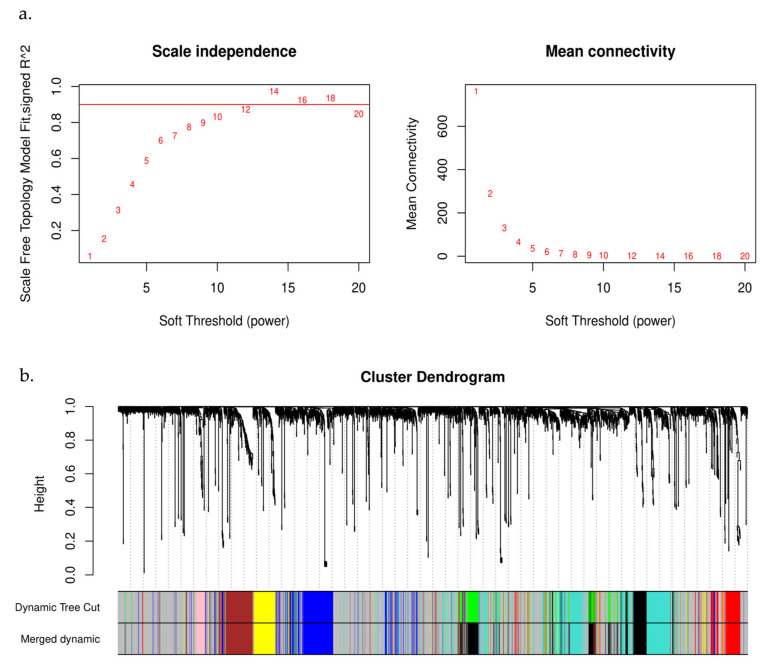
WGCNA analysis. (**a**) The soft threshold selection, including the analysis of the scale-free topology fitting index R2 (left) and the mean connectivity for various soft threshold powers (right). In the left panel, the red line indicates that R2 = 5. (**b**) A clustering diagram of the gene modules denoted by distinct colors. The gene dendrogram was obtained by dissimilarity clustering, with the colors of the corresponding modules represented by colored lines, based on the consensus topological overlap. Each colored line represents a color-coded module containing a set of highly connected genes.

**Figure 3 biomedicines-11-00518-f003:**
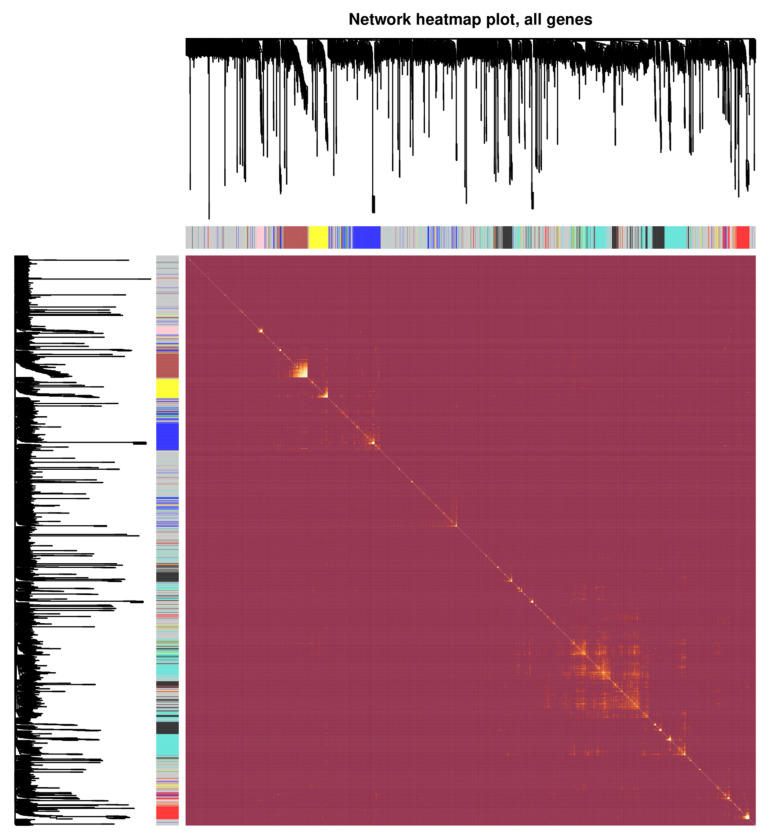
Visualization of the WGCNA network. The heatmap plot represents the gene network. The heatmap illustrates the topological overlap matrix among all genes in the analysis.

**Figure 4 biomedicines-11-00518-f004:**
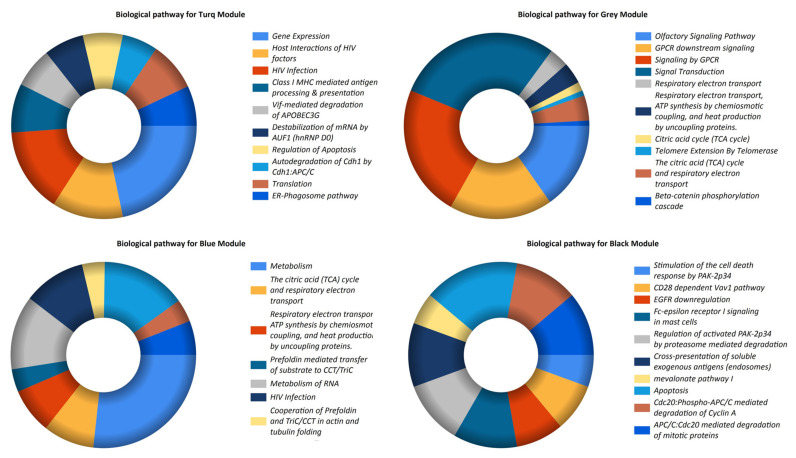
Biological pathway enrichment analysis of genes in the co-expression modules. Biological pathway enrichment analysis of the genes was performed using FunRich software.

**Figure 5 biomedicines-11-00518-f005:**
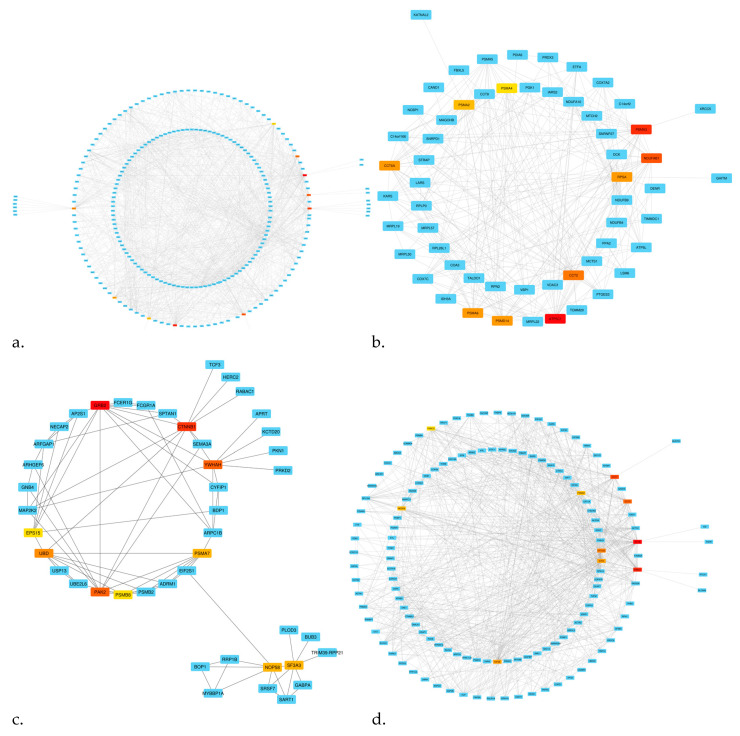
(**a**–**d**) The gray, blue, black, and turquoise modules forming the PPI network. Rectangles and lines represent genes and the interactions of proteins between genes, respectively. The hub genes are colored red, orange, and yellow according to their score, from highest to lowest.

**Table 1 biomedicines-11-00518-t001:** Detail of nine dysregulated lncRNAs.

LncRNA	LogFC	*p*-Value	Expression	Co-Expression Module	PPI Networks	Hub Genes
*LOC102725104*	1.825633999	0.005721501	Upregulated	Turquoise	134 node and 842 edge	*ACTB*, *GNB2L1*, *CCT7*, *CCT3*, *EFTUD2*, *EIF3B*, *EIF3I*, *NEDD8*, *PSMD2*, *PSMC5*
*LINC00328*	1.722041383	0.014404303				
*LOC102725070*	1.629786576	0.01119095				
*LOC101059935*	1.621249989	0.004591495				
*LOC100127951*	1.577888412	0.000306974	Upregulated	Gray	238 node and 1142 edge	*HSP90AA1*, *GART*, *ESR1*, *RPS3*, *JUN*, *POLR2C*, *ALDH18A1*, *RUVBL1*, *TPI1*, *NHP2*
*LOC648987*	−1.222919534	0.005645454	Downregulated			
*LOC93622*	−1.24184329	0.006655765	Downregulated			
*LOC644936*	−1.288710709	0.001962644	Downregulated	Black	44 node and 85 edge	*GRB2*, *CTNNB1*, *YWHAH*, *PAK2*, *UBD*, *PSMA7*, *NOP58*, *SF3A3*, *PSMB8*, *EPS15*
*LINC00998*	−1.383262604	0.012944269	Downregulated	Blue	58 node and 296 edge	*ATP5C1*, *PSMA3*, *NDUFAB1*, *CCT2*, *PSMA6*, *CCT6A*, *PSMD14*, *RPSA*, *PSMA2*, *PSMA4*

**Table 2 biomedicines-11-00518-t002:** Pathway enrichments of the co-expressed modules.

LncRNAs	Co-Expressed Modules	Pathways
*LOC102725104* *LINC00328* *LOC102725070* *LOC101059935*	Turquoise	Gene expression
		Host interactions of HIV factors
		HIV infection
		Class I MHC mediated antigen processing and presentation
		Vif-mediated degradation of APOBEC3G
		Destabilization of mRNA by AUF1 (hnRNP D0)
		Regulation of apoptosis
		Autodegradation of Cdh1 by Cdh1: APC/C
		Translation
		ER–phagosome pathway
*LOC100127951* *LOC648987* *LOC93622*	Gray	Olfactory signaling pathway
		GPCR downstream signaling
		Signaling by GPCR
		Signal transduction
		Respiratory electron transport
		Respiratory electron transport, ATP synthesis by chemiosmotic coupling, and heat production by uncoupling proteins.
		Citric acid cycle (TCA cycle)
		Telomere extension by telomerase
		The citric acid (TCA) cycle and respiratory electron transport
		Beta-catenin phosphorylation cascade
*LINC00998*	Blue	Metabolism
		The citric acid (TCA) cycle and respiratory electron transport
		Respiratory electron transport, ATP synthesis by chemiosmotic coupling, and heat production by uncoupling proteins.
		Prefoldin mediated transfer of substrate to CCT/TriC
		Metabolism of RNA
		HIV infection
		Cooperation of prefoldin and TriC/CCT in actin and tubulin folding
		Gene expression
		Chaperonin-mediated protein folding
		Autodegradation of Cdh1 by Cdh1: APC/C
*LOC644936*	Black	Stimulation of the cell death response by PAK-2p34
		CD28-dependent Vav1 pathway
		EGFR downregulation
		Fc-epsilon receptor I signaling in mast cells
		Regulation of activated PAK-2p34 by proteasome mediated degradation
		Cross-presentation of soluble exogenous antigens (endosomes)
		Mevalonate pathway I
		Apoptosis
		Cdc20:Phospho-APC/C mediated degradation of Cyclin A
		APC/C:Cdc20 mediated degradation of mitotic proteins

## Data Availability

Not applicable.

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
