# Peer review of "Weighted Gene Co-Expression Network Analysis (WGCNA) Discovered Novel Long Non-Coding RNAs for Polycystic Ovary Syndrome"

_biomedicines, 2023, doi:10.3390/biomedicines11020518_

Round 1

Reviewer 1 Report

A review report of “The Weighted Correlation Network Analysis Discovered Novel Long Non-Coding RNAs for Polycystic Ovary Syndrome” by Heidarzadehpilehrood et al

Heidarzadehpilehrood et al. examined PCOS microarray data from the GEO database for lncRNAs to which PCOS pathology may be attributable by WGCNA. They found nine candidate lncRNAs, though no additional experimental data to support their conclusions was provided. Although the manuscript appears to contain useful information, extensive modifications would be needed before publishing.

Comments

1. Lines 24-25: The phrase “turquoise, grey, blue, and black co-expression modules” does not make sense.

2. Line 81: Describe what WGCNA stands for.

3. Line 91: It should be described that the 29 samples were from FACS-isolated four different cell types. Similar WGCNA for each cell type is interesting, and may provide quite different results.

4. It would be helpful to describe characteristics of the nine lncRNAs such as length and interspecies homology.

5. Much more description in the “Results” section and the figure legends would be needed for readers to better understand the contents.

6. Lines 514-515: A journal name is missing.

7. Figure 5: Higher resolution would be needed.

Author Response

[Cover Letter]
Dear Prof. Dr. Shaker A. Mousa,
Editor-in-Chief
Dear Editors,
We appreciate you taking the time to evaluate our paper and provide useful feedback. The authors have read every remark and sought to respond to each one. We hope that the information meets your high quality and professionalism criteria. If you have any further constructive feedback, please let the authors know.
Here are the points addressed.
Best Regards,
Correspondent author
Dr. Habibah Abdul Hamid
Associate Professor
Department of Obstetrics & Gynaecology, Faculty of Medicine and Health Sciences, Universiti Putra Malaysia, Selangor, Malaysia
Email: [email protected]
Tel: 603 - 8947 2640
Response to reviewer 1 Comments
Comment 1. Lines 24-25: The phrase “turquoise, grey, blue, and black co-expression modules” does not make sense.
Comment 2. Line 81: Describe what WGCNA stands for. Answer to comments 1 and 2:
Thank you very much for the reminder. We have made revisions accordingly. The modifications have been applied [Page 1, Lines 26-30] and [Page 2 and 3, Lines 90-101].
Comment 3. Line 91: It should be described that the 29 samples were from FACS-isolated four different cell types. Similar WGCNA for each cell type is interesting, and may provide quite different results. Answer to comment 3:
Thank you very much for your nice reminders that helped us improve this sentences. We have made revisions accordingly [Page 3, Lines 112-115].
(Note: About WGCNA sample size: WGCNA recommends at least 15 samples in order to build reliable networks. In a typical high-throughput setting, correlations on fewer than 15 samples will simply be too noisy for the network to be biologically meaningful. Hence, we did not perform WGCNA in four different cell types separately because the number of samples in each subtype was below 15.
Comment 4. It would be helpful to describe characteristics of the nine lncRNAs such as length and interspecies homology. Answer to comment 4:
Thank you very much for the reminder. To the best of our knowledge, some of these lncRNAs have been evaluated in other diseases; however, none of these lncRNAs have been studied in PCOS. Also, the majority of these lncRNAs are new and have been recently investigated in other diseases, and a limited/rare number of functional studies are available about them. Hence, we try to do our best and add more researches about these nine novel LncRNAs to the updated version of the manuscript [Page 13, Lines 245-284].
Comment 5. Much more description in the “Results” section and the figure legends would be needed for readers to better understand the contents. Answer to comment 5:
Thanks for your kind reminders. The error has been corrected in Results and Figure and Legends.
Comment 6. Lines 514-515: A journal name is missing. Answer to comment 6:
Thank you very much for the reminder. Also, we went through the entire Reference Section in order to eliminate wrong references and also update/add new references.
Comment 7. Figure 5: Higher resolution would be needed
Answer to comment 7:
Thank you very much. The quality of All Figures are improved in the updated version of the manuscript. Writer's Notes to Editors:
Dear Editors, in addition to your valuable comments, we also went through the entire manuscript to eliminate grammatical mistakes and also updated the manuscripts in different aspects.

Reviewer 2 Report

This study investigated changes in long non-coding RNAs and their correlation in network analysis in the context of PCOS. For this bioinformatic study, the gene expression of GSE48301 comprising PCOS patient and normal control tissue samples was evaluated using the R limma package . First, WGCNA analysis was used to determine the co-expression genes of lncRNAs. Subsequently, gene identification and pathway enrichment analysis were used.  The results obtained through these approaches indicate enrichment in relevant pathways to PCOS, including gene expression, signal transuciton metbaolism, and apoptosis. The study uncovered nine novel PCOS_related lncRNAs. The translational or functional significance of these genes was not addressed in this study,

Their functional significance needs to be assessed in future study

Author Response

[Cover Letter]
Dear Prof. Dr. Shaker A. Mousa,
Editor-in-Chief
Dear Editors,
We appreciate you taking the time to evaluate our paper and provide useful feedback. The authors have read every remark and sought to respond to each one. We hope that the information meets your high quality and professionalism criteria. If you have any further constructive feedback, please let the authors know.
Here are the points addressed.
Best Regards,
Correspondent author
Dr. Habibah Abdul Hamid
Associate Professor
Department of Obstetrics & Gynaecology, Faculty of Medicine and Health Sciences, Universiti Putra Malaysia, Selangor, Malaysia
Email: [email protected]
Tel: 603 - 8947 2640
Response to reviewer 2 Comments
Thank you very much for the careful reading of our manuscript. We went through the entire manuscript to eliminate grammatical mistakes and also updated the manuscripts in different aspects.

Round 2

Reviewer 1 Report

A second review report of “The Weighted Correlation Network Analysis Discovered Novel Long Non-Coding RNAs for Polycystic Ovary Syndrome” by Heidarzadehpilehrood et al

The manuscript has been well modified, so that it should be accepted for publication in Biomedicines.